# Variation in Climate Signals in Teak Tree-Ring Chronologies in Two Different Growth Areas

**Sineenart Preechamart [1], Nathsuda Pumijumnong [1,\*], Paramate Payomrat [2] and Supaporn Buajan [2]** 

[1] Faculty of Environment and Resource Studies, Mahidol University, Nakhon Pathom 73170, Thailand; sineenart.pre@student.mahidol.ac.th
[2] Tree-Ring and Climate Change Research Center, Faculty of Environment and Resource Studies, Mahidol University, Nakhon Pathom 73170, Thailand; paramate.ohm@gmail.com (P.P.); buajan_s@hotmail.com (S.B.)
[\*] Correspondence: nathsuda.pum@mahidol.ac.th; Tel.: +66-244-15000 (ext. 2311)

**Abstract:** We developed two tree-ring chronologies of teak (*Tectona grandis* L.f.) from Mae Tuen (462-year, 1555–2016) and Umphang (165-year, 1852–2016) in Tak province, northwestern Thailand. The chronologies were based on 67 and 71 living teak trees, respectively. We used crossdating methods to check and verify the tree-ring width data and tree-ring chronology construction using the ARSTAN program. In this study, the two teak tree-ring chronologies from two different growth areas could not be crossdated. The relationship among these chronologies is, thus, relatively low ($r = 0.33$, $n = 165$, $p < 0.01$). This result shows that the growth of tree-ring structure from two sites can be affected by a variety of non-climatic patterns due to site variation, such as topography, nutrient, light, and internal factors. However, these chronologies have a significant positive correlation with rainfall, during the pre-monsoon season (April to May). As demonstrated by the spatial correlation patterns, these chronologies represent April to May rainfall, which was a limiting factor of teak growth from northwestern Thailand. While the difference in surface temperatures of the Indian Ocean Dipole (IOD) might not be affected by rainfall, its unstable relationship with the El Niño-Southern Oscillation (ENSO) was noted to have occurred.

**Keywords:** dendroclimatology; ring-width; teak (*Tectona grandis* L.f.)

## 1. Introduction

The Asian monsoon system (AMS) is a key variable for global climate change and plays a significant role in large-scale climate variability [1]. The AMS is primarily driven by convective, radiative, and sensible heat sources/sinks [2,3]. Southwest monsoons are formed due to intense low pressure systems formed over the Tibetan plateau. Northeast monsoons are associated with high pressure cells over the Tibetan and Siberian plateaus. Countries like India, Indonesia, Bangladesh, Myanmar, and Thailand receive most of the annual rainfall during the southwest monsoon season [4]. AMS is composed of three inter-linked components: Indian summer monsoon (ISM), East Asian summer monsoon (EASM), and Southeast Asian summer monsoon (SASM) [5]. Studies on the past AMS have sought to explain the ISM or EASM and the interaction between the two monsoons [6–8]. In addition, recent studies have focused on the influence of the El Niño phenomena and its effect on the amount of rainfall or drought in Southeast Asia [9,10]. The interaction between the El Niño phenomenon and the variation in sea surface temperatures in the Indian Ocean influences the amount of rainfall that will fall on land [11–13]. In contrast, currently, there are not many studies on the SASM. Indeed, to better understand the influence of climate in Southeast Asian countries, long climate information and extensive weather stations coverage area are required. These are limitations of the

study and the understanding of the climate in Southeast Asia. Further study of these issues needs to be encouraged. Due to the lack of extensive coverage of weather stations across Southeast Asia, the available weather information is relatively limited. Past climate data was derived from proxies, such as lake sediment, coral reef, pollen, speleothem, archaeological material, and tree-ring records [14,15]. In comparison with other proxies, tree-rings have proven to be an annually resolved and precisely dated proxy [14,16]. Teak (*Tectona grandis* L.f.), which normally develops clear annual rings, is a native of Southeast Asia. It is a good proxy that can be used to represent climatic characteristics [17–21]. This study uses teak that is grown in northwestern Thailand, which is near the border with Myanmar and is expected to gain moisture directly from the Indian Ocean. Regional tree-ring research over the past few decades has demonstrated that quantitative reconstruction of past monsoonal climate over the period of few centuries would be possible through dendroclimatology [15]. However, it is unclear whether the main source of rainfall in Thailand comes from the Indian Ocean or from the South China Sea, and the influence of the El Niño phenomenon and the surface temperatures of the ocean are unknown. The role that rainfall plays in the climatic occurrences within the country will need to be determined. Thus, for this study, we use the teak ring widths from two different sites; Mae Tuen (MT) and Umphang (UP) in Tak province, northwestern Thailand to examine the sources of rainfall that contribute to tree growth. Furthermore, we examine the teak tree-ring chronology's response to the Self-Calibrating Palmer Drought Severity Index (scPDSI), El Niño-Southern Oscillation (ENSO), and the difference in surface temperatures of the Indian Ocean Dipole (IOD).

## 2. Materials and Methods

### 2.1. Study Area

This study has two study sites; Site 1: Mae Teun (MT) site is situated in Mae Teun Wildlife Sanctuary, Tak province, northwestern Thailand at an elevation from 200 to 1200 m a.s.l. The geomorphology is shale rock and well-drained soils. The slope of MT site is moderate to steep, and there are complex mountains parallel to Mae Tuen River that are covered with mixed deciduous forest. The MT site was separated into 2 sub-sites; Mae Tuen (MTT) (17°07′ N, 98°39′ E) and Ceva (CVT) (17°08′ N, 98°37′ E). The distance between MT sub-sites is approximately 14 km. Site 2: Umphang (UP) site is situated in Umphang Wildlife Sanctuary, Tak province, which is approximately 130 km southwest from MT site. The UP site was separated into 3 sub-sites; Pha Lueat (PLT) (15°58′ N, 98°48′ E), Tasai (TST) (15°56′ N, 98°48′ E), and Palatha (PLTT) (15°50 N, 98°50′ E). The distance between PLT–TST is approximately 4 km and between TST–PLTT is approximately 12 km. In this area, the geographical features of the Umphang Wildlife Sanctuary consist of extensive and high mountains, however, the slope is lower than the MT site (Figure 1a).

The seasonal climate of the study areas is under the influence of two monsoons of seasonal character, i.e., southwest monsoon (summer monsoon) and northeast monsoon (winter monsoon). The southwest monsoon brings warm moist air from the Indian Ocean. The rainy season in northwestern Thailand occurs between May and October. According to the longest period of meteorological data recorded (the period of 1951–2016) by the Mae Sot Meteorological Station in Tak province, which is the nearest vicinity to both study sites (MT and UP), the highest rainfall occurred in August, and the lowest rainfall occurred in December. The total annual rainfall was 1493.43 mm, and the annual average temperature was 26.19 °C. During the rainy season, mean monthly rainfall ranged from 100 to 355 mm (mean = 235 mm), mean monthly temperatures ranged from 24.70 to 30.86 °C (seasonal mean = 26.93 °C), and relative humidity ranged from 55% to 90%, (mean = 82.4%) (Figure 1b,c). However, the local climate at the UP site from The Umphang Meteorological Station, which is the nearest to the UP site, the highest rainfall occurred in September, and lowest rainfall occurs in December. The total annual rainfall is 1468.65 mm, and the annual average temperature is 24.09 °C. During the rainy season, mean monthly rainfall ranged from 151 to 257 mm (mean = 211 mm),

mean monthly temperatures ranged from 24.59 to 26.35 °C (seasonal mean = 25.16 °C), and relative humidity ranged from 68 to 98% (mean = 84.8%) (Figure 1b).

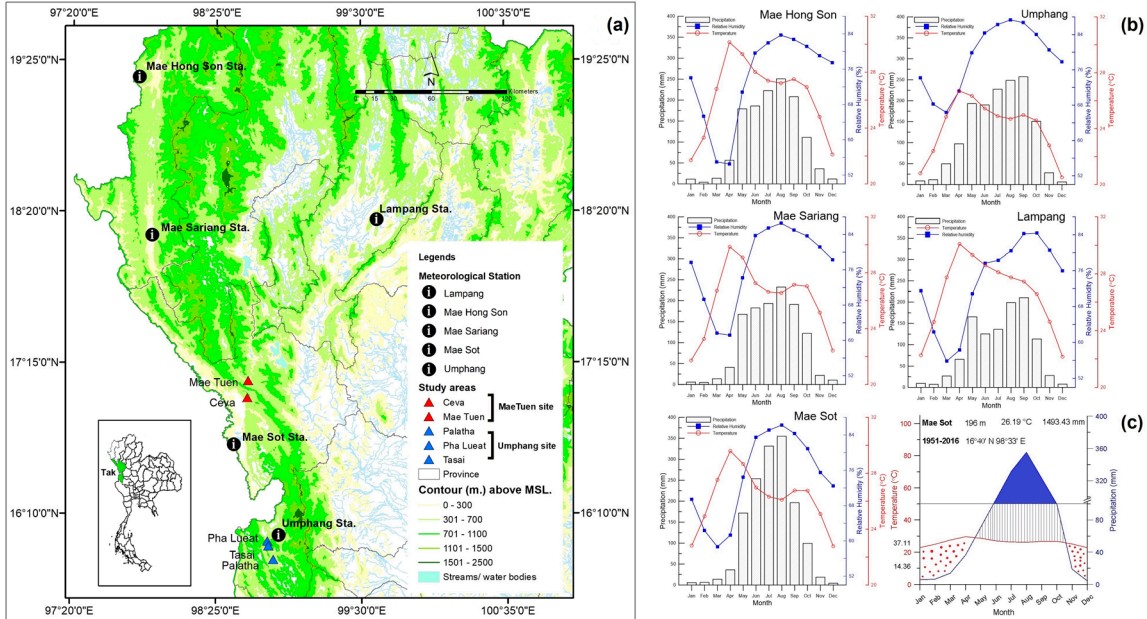

**Figure 1.** (**a**) Map of the study areas, Mae Tuen (MT) (red triangles) and Umphang (UP) (blue triangles) and the meteorological stations (black circles). (**b**) Mean climate conditions; mean monthly rainfall (gray vertical bars), temperature (red line with circles), and relative humidity (blue line with filled squares) in the period of 1951–2016 from Mae Hong Son (19°18′ N, 97°50′ E), 1951–2016 from Mae Sariang (18°10′ N, 97°56′ E), 1951–2016 from Mae Sot (16°40′ N, 98°33′ E), 1977–2016 from Umphang (16°01′ N, 98°51′ E), and 1951–2016 from Lampang (18°17′ N, 99°11′ E) Meteorological Station. (**c**) Walter and Lieth climate diagram; mean monthly rainfall (blue line) and temperature (red line) from Mae Sot Meteorological Station (the period of 1951–2016), dry period (red dot area), humid period (blue line area), and wet period (blue area).

## 2.2. Sample Collection and Preparation

The samples were collected from healthy and old teak trees, which the diameter at the breast height (1.30 m above the ground) ranged from 150 to 340 cm. For each tree, 5-mm-increment borers were used to collect two or more sample in the opposite direction. A total of 362 samples (183 trees) were taken in the years 1999, 2008, and 2017 (Table 1).

**Table 1.** Detail of sample data from Mae Tuen (MT) and Umphang (UP) in Tak province, northwestern Thailand.

| Site | | Number of Samples | | Diameter at Breast Height (DBH) (cm) | Meters Above Mean Sea Level (MASL) | Average Age of Trees |
|---|---|---|---|---|---|---|
| Sub-site | Location | Trees | Samples | | | |
| **Mae Tuen Site** | | **85** | **166** | | | |
| Mae Tuen (MTT) | 17°07′ N, 98°39′ E | 75 | 146 | 195–270 | 390–530 | 207 |
| Ceva (CVT) | 17°08′ N, 98°37′ E | 10 | 20 | 150–300 | 360–480 | 116 |
| **Umphang Site** | | **98** | **196** | | | |
| Pha Lueat (PLT) | 15°58′ N, 98°48′ E | 11 | 23 | 180–340 | 440–500 | 115 |
| Tasai (TST) | 15°56′ N, 98°48′ E | 72 | 143 | 195–310 | 480–520 | 106 |
| Palatha (PLTT) | 15°50′ N, 98°50′ E | 15 | 30 | 190–305 | 440–460 | 111 |
| **Total** | | **183** | **362** | | | |

All of the cores, which were removed with an increment borer, were air dried before mounting. When the cores were sufficiently dry, they were glued onto a supporting wood. The most important aspect of the mounting was the correct alignment of the cells in the core. If the core was twisted when

it was extracted, it must be twisted back to the correct alignment. Cores were fixed to the mount using water-soluble glue so that they could be realigned if necessary. After that, they were clamped to the core to hold them in place with adhesive tape while the glue set. The cores were stored this way until the glue dried. The core surfaces were prepared for study by sanding the cores with a rotary-tool or by hand with sandpaper until there was a clear view of the annual rings [22,23].

*2.3. Tree-Ring Measurement and Tree-Ring Chronology Construction*

The samples were measured using a moving stage (LINTAB Digital linear positioning table by Frank Rinn Company, Heidelberg, German) and microscope (Leica GZ6 microscope) that was interfaced with a computer running program TSAP-Win™, version 4.64 for Microsoft Windows, which served as the data recorder and editor. Crossdating involves matching tree-ring width patterns from different samples using the TSAP-Win™ program [24]. Samples were dated to the calendar year of the tree-ring formation and crossdated. At first, tree-ring width series were crossdated between cores from the same tree and then, crossdated between trees from the same sub-sites. The sub-site chronologies were crossdated separately for each site (MT and UP), and MT and UP chronologies were crossdated among sites. The accuracy of crossdating was subsequently checked by using the COFECHA program (version 3.02P) [25]. Finally, residual chronologies that were constructed using the ARSTAN program [26] were used for fitting a 66-year spline function to each ring series to eliminate the age trend. Autoregressive modeling was applied, and the series were averaged, using the robust mean for the white noise residual chronology [27].

*2.4. Climate Data and Statistical Analyses*

The chronology signal strength was quantified using mean series inter-correlation (Rbar) [23] and the expressed population signal (EPS) [28], which indicates how well the ring-width estimates a theoretically infinite population. Local climatic data (total monthly rainfall, mean monthly temperature, and mean monthly relative humidity) obtained from five local meteorological stations, Thai Meteorological Department (Table 2), and regional climate data (total monthly rainfall, mean monthly temperature, and mean monthly relative humidity) from the Climatic Research Unit time-series (CRU TS) 4.01 [29] gridded data-sets (14°–19° N, 98°–101° E) from 1951–2016 with resolutions of 0.5° × 0.5° were used to evaluate the climate effects of the ring-width data.

**Table 2.** Detail of five local meteorological stations.

| Station | Location | Distance (km) | | Period of Data |
|---|---|---|---|---|
| | | UP Site | MT Site | |
| Mae Hong Son (MHS) | 19°18′ N, 97°50′ E | 390 | 260 | 1951–2016 |
| Mae Sariang (MSR) | 18°10′ N, 97°56′ E | 270 | 140 | 1951–2016 |
| Mae Sot (MS) | 16°40′ N, 98°33′ E | 90 | 45 | 1951–2016 |
| Umphang (UP) | 16°01′ N, 98°51′ E | 15 | 120 | 1977–2016 |
| Lampang (LP) | 18°17′ N, 99°11′ E | 275 | 170 | 1951–2016 |

In addition, this study tested the similarity with various climatic indices: The Dipole mode index (DMI) was used to investigate the influence of the Indian monsoon on the temperature over northwestern Thailand [30], Kaplan Niño 3, 3.4, and 4 indices were used to test the stability of the teleconnection between the tropical Pacific sea surface temperatures (SSTs), and the Self-calibrating Palmer Drought Severity Index (scPDSI), during the period of 1951–2016. The correlation between tree-ring chronologies and monthly climatic data and other parameters was tested using SPSS version 16.0 and calculated by Pearson's correlation coefficient (*r*).

## 3. Result

### 3.1. Tree-Ring Chronology

In this study, a total of 112 ring-width MT series (67 trees) and 101 ring-width UP series (71 trees) of teak were successfully crossdated and used to create the MT and UP tree-ring chronologies. The MT and UP chronologies span the period of 1555–2016 (462 years) and 1852–2016 (165 years), respectively. The results of some descriptive statistics of tree-ring chronology from each site of the COFECHA program are illustrated in Table 3. The teak chronologies and number of samples through time are presented in parallel with the running EPS and Rbar, which were calculated using 40-year moving windows with 20-year overlapping (Figure 2).

**Table 3.** Statistics of tree-ring chronologies from two different sites in Tak province.

| Site | Mae Tuen (MT) | Umphang (UP) |
|---|---|---|
| Master series | 1555–2016 (462 years) | 1852–2016 (165 years) |
| No. of trees (cores) | 67 (112) | 71 (101) |
| Meters above mean sea level (MASL) | 350–530 | 430–520 |
| Correlation with master | 0.451 | 0.458 |
| Standard deviation (SD) | 0.113 | 0.156 |
| Autocorrelation | 0.675 | 0.693 |
| Mean sensitivity | 0.393 | 0.345 |

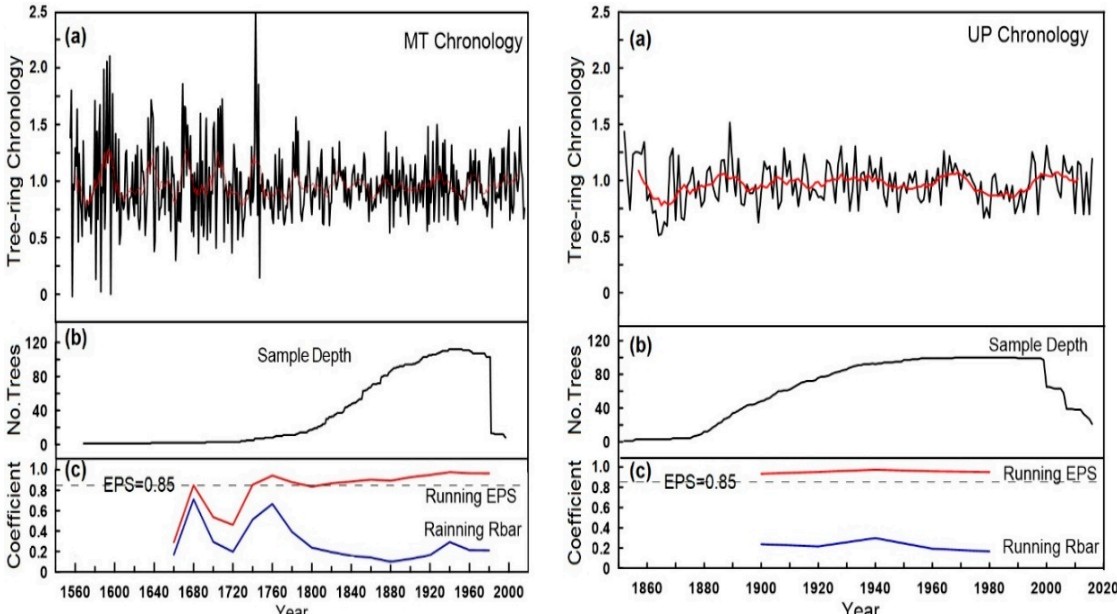

**Figure 2.** (**a**) The MT and UP chronologies, (**b**) variation in tree samples size, and (**c**) variation in running expressed population signal (EPS) and mean series inter-correlation (Rbar).

To date, crossdating the two chronologies in an indexed format could not be matched. Furthermore, correlation coefficient between MT and UP chronologies over the period from 1852–2016 (165 years) showed relatively low significant correlations (Pearson; $r = 0.33$, $n = 165$, $p < 0.01$) (Figure 3).

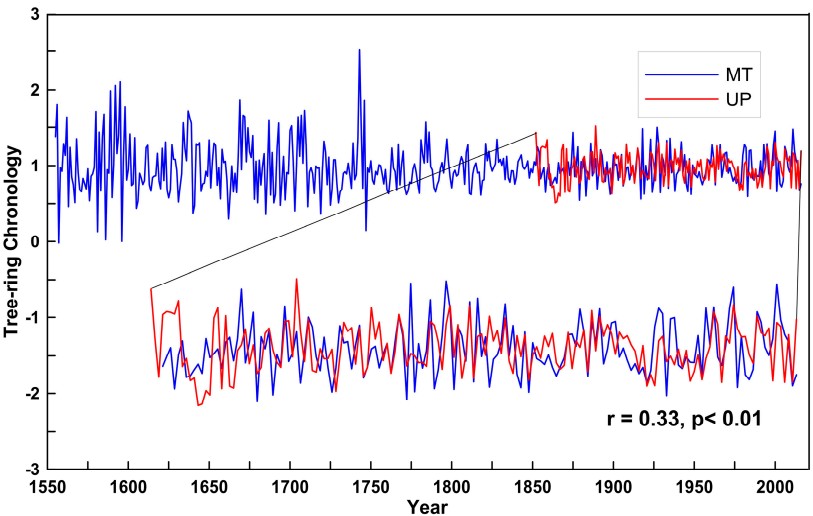

**Figure 3.** Correlation coefficients among teak tree-ring chronologies.

### 3.2. Tree-Ring Chronologies and Climate Relationship

Correlation analysis was carried out on climatic factors (i.e., total monthly rainfall, mean monthly temperature, and mean monthly relative humidity) from local meteorological stations and at a regional scale (CRU TS 4.01). Among the two areas, MT teak chronology shows significant positive correlation with rainfall from local meteorological station during the months April–May and July on pre-monsoon season; April (MHS, $r = 0.33$; UP, $r = 0.32$) and May (MHS, $r = 0.32$; MSR, $r = 0.38$; MS, $r = 0.36$; LP, $r = 0.43$), and July (MHS, $r = 0.26$), April–May regional rainfall data from CRU TS 4.01 ($r = 0.36$ and 0.38, respectively) (Figure 4) and shows significant positive correlation with relative humidity from May to June ($r = 0.29–0.33$). Conversely, this chronology shows a significant negative correlation with temperature from local meteorological stations in the months April–June and August ($r = -0.45$ to $-0.27$), May and July–September regional temperature from CRU TS 4.01 ($r = -0.39$ to $-0.26$) (Figure 4). The MT chronology showed a significant positive correlation with the mean and variance adjusted average of the four scPDSI grid point that were used to cover the study area from May to September ($r = 0.25–0.28$), with the strongest being in May ($r = 0.28$, $n = 66$, $p < 0.05$).

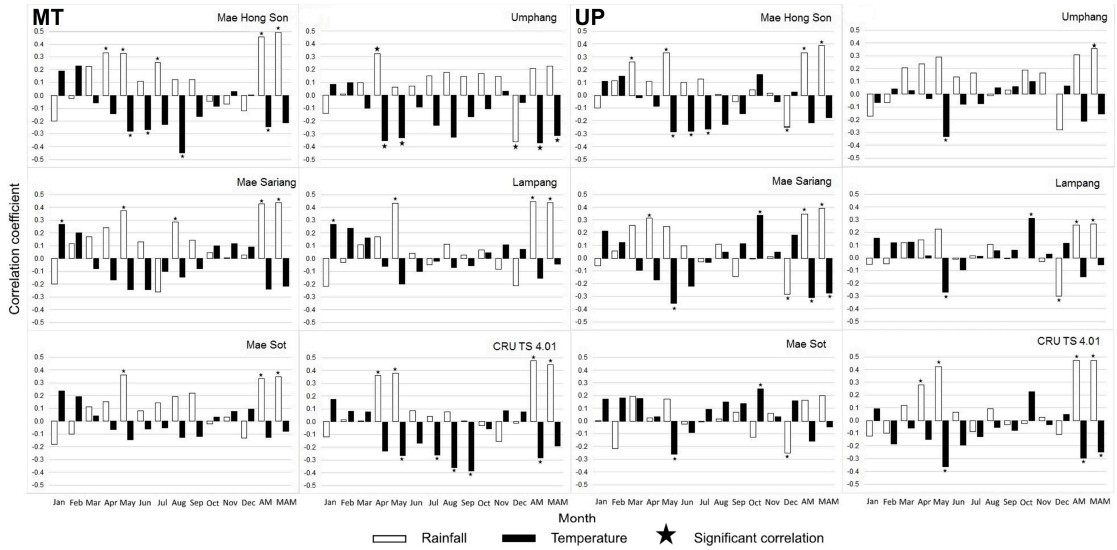

**Figure 4.** Correlations between tree-ring chronology and some climatic data (rainfall (white vertical bars) and temperature (black vertical bars)) from five meteorological stations in northwestern Thailand and the CRU TS 4.01 data set (1951–2016) at the 95% significance confidence level.

UP teak residual chronology showed significant positive correlation with rainfall from the local meteorological station during March, and respectively May; March (MHS, *r* =0.26), April (MSR, *r* = 0.32), May (MHS, *r* = 0.34), April–May regional rainfall data from CRU TS 4.01 (*r* = 0.28 and 0.42, respectively) (Figure 4), and showed significant positive correlation with relative humidity from March to June, which was strongest in May (*r* = 0.25–0.40). In contrast, the UP chronology showed a significant negative correlation with temperature from local meteorological stations from May to July, that was the strongest in May, and May regional temperature from CRU TS 4.01 (*r* = −0.36) (Figure 4). UP chronology showed significant positive correlation with scPDSI from January, and March to September (*r* = 0.25–0.44), strongest in May (*r* = 0.44, *n* = 66, *p* < 0.01).

The correlation between the monthly data sets with both tree-ring chronologies for rainfall and temperature was determined. Correlation of April–May (AM) and March–May (MAM) monthly data sets showed the two highest correlations, illustrated in Figure 4 and Table 4. To show the special representation, spatial correlation analysis between tree-ring chronology and CRU TS 4.01 April–May rainfall, which had the strongest relationship among the other climate signals, were calculated during the period of 1951–2016 (Figure 5). The correlation showed high significance between tree-ring chronology and CRU TS 4.01 April–May rainfall over western and part of northern and upper-central Thailand.

**Table 4.** Correlation coefficients between tree-ring data and rainfall and temperature.

| Period | MT Chronology | | | | | | UP Chronology | | | | | |
|---|---|---|---|---|---|---|---|---|---|---|---|---|
| | MHS | MSR | MS | UP | LP | CRU | MHS | MSR | MS | UP | LP | CRU |
| | Rain/ Temp. | Rain/ Temp. | Rain/ Temp. | Rain/ Temp. | Rain/ Temp. | Rain/ Temp. | Rain/ Temp. | Rain/ Temp. | Rain/ Temp. | Rain/ Temp. | Rain/ Temp. | Rain/ Temp. |
| AM | 0.46 ** −0.25* | 0.43 ** −0.24 | 0.33 ** −0.13 | 0.21 −0.37 * | 0.44 ** −0.15 | 0.47 ** −0.29 * | 0.34 ** −0.22 | 0.35 ** −0.31 ** | 0.16 −0.16 | 0.31 −0.21 | 0.26 * −0.15 | 0.47 ** −0.30 * |
| MAM | 0.49 ** −0.21 | 0.44 ** −0.22 | 0.35 ** −0.08 | 0.23 −0.31 | 0.44 ** −0.04 | 0.44 ** −0.19 | 0.39 ** −0.17 | 0.39 ** −0.28 * | 0.20 −0.05 | 0.36 * −0.15 | 0.27 * −0.06 | 0.47 ** −0.25 * |

Monthly data set; AM = April–May, MAM = March–May; Rain = Total monthly rainfall, Temp. = Mean monthly temperature; Meteorological Station: MHS = Mae Hong Son Station; period of record 1951–2016 rainfall and temperature; MSR = Mae Sariang Station, period of record 1951–2016 rainfall, and 1961–2016 temperature; MS = Mae Sot Station; period of record 1951–2016 rainfall and temperature; UP = Umphang Station, period of record 1977–2016 rainfall and 1980–2016 temperature; LP = Lampang Station, period of record 1951–2016 rainfall and temperature. ** Correlation is significant at the 0.01 level, * Correlation is significant at the 0.05 level.

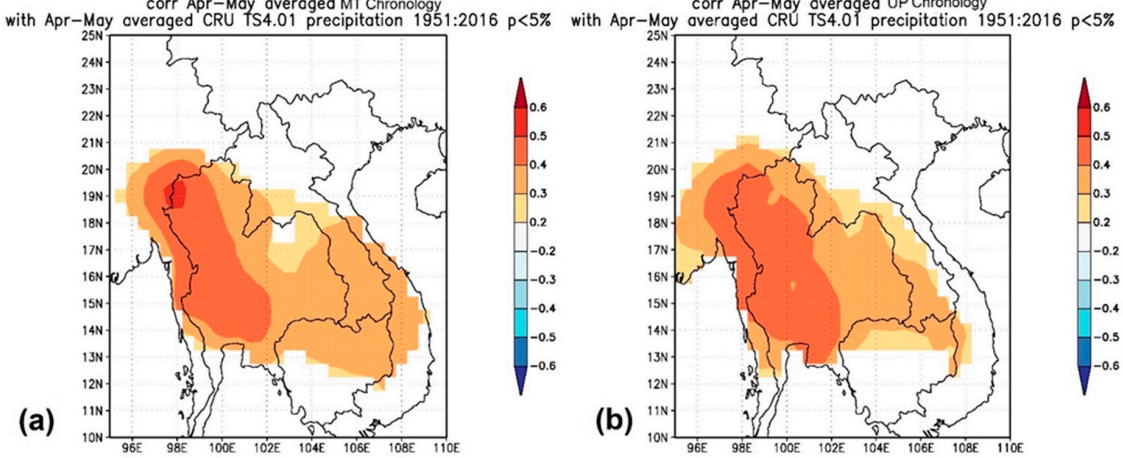

**Figure 5.** Spatial correlations between tree-ring chronology and April–May rainfall, obtained from CRU TS 4.01 data set for the period 1951–2016. (**a**) MT chronology. (**b**) UP chronology.

*3.3. Teleconnection with Climate Driving Force*

These chronologies showed no significant correlation with Dipole mode index (DMI). Pearson's correlation showed low (*r* = −0.17 to −0.20, *n* = 147, *p* < 0.05) but significant negative correlation

between UP chronology and Kaplan Niño indices (1870–2016); Niño 3 (July–December), Niño 3.4 (July–August, and December), and Niño 4 (July–August and November–December), but showed no significant correlation with MT chronology. To investigate the changes in the long-term relationship between tree-ring chronologies, and the Niño 3.4 index, a 21-year running correlation was performed (Figure 6). It was found that the 21-year running correlation between UP and MT chronologies and Niño 3.4 index exhibited the same unstable correlation trends. MT and UP chronologies showed relatively low correlation with Niño 3.4 in the period of 1870–1980, except MT in the period 1905–1910 and 1965–1975, both of which showed strong positive correlations with Niño 3.4. Particularly, these chronologies showed a high negative correlation with Niño 3.4 after 1980.

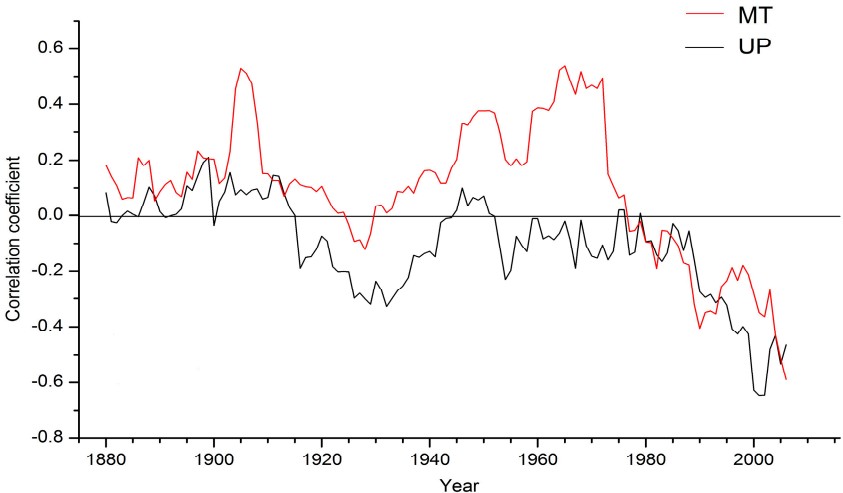

**Figure 6.** Twenty-one-year running correlation between tree-ring chronologies and April–May Niño 3.4.

## 4. Discussion

### 4.1. Tree-Ring Growth Response to Climate

The tree-ring chronologies were correlated with seasonal climatic variables to examine the climatic signal in tree growth. The teak chronologies showed a positive correlation with rainfall and relative humidity during the same period. The MT and UP chronologies exhibited the strongest correlation in the April–May monthly rainfall data set. The climate-growth relationship suggested that the growth of teak in this particular study area was mainly controlled by the early monsoon rainfall. The amount of rainfall from the transitional monsoon months (March–July) could thus be counted as a critical factor that influenced teak growth in Thailand [21]. Spatial correlation between the tree-ring chronologies and April–May rainfall (Figure 5.) showed that the tree-ring chronology was related to pre-monsoon rainfall in the western parts, along with parts of northern and upper-central Thailand. Therefore, the tree-ring chronologies are a reliable proxy for large-scale rainfall. These results agree with previous dendroclimatological studies from teak in Thailand that discovered a significant positive relationship between rainfall and teak growth in the monsoon season [19,21,31–33]. This was supported by the observation of cambial activity in Thai teak that started in the second half of April [32], a transition period between the dry and wet season in Thailand. In addition, observation of teak growth using dendrometer bands in northern Thailand showed the importance of rainfall in the transitional period for teak growth [19,34]. When compared with several studies in the Asian monsoon countries, Thai teak responded to rainfall earlier than teak in central India (June–September) that capture signals of post-monsoon rainfall [35], and Indonesia (June–September) [36] and Myanmar (April–August) [37,38] that respond to a long period of rainfall. Both chronologies had a significant positive correlation with relative humidity, the highest being in May due to the humidity variable being dependent on the amount of rainfall. This differed from the study of Indian teak that showed

that the moisture availability in the post-monsoon period of the previous year had a significant role in the development of teak growth in the current year [39].

In the case of temperature, the correlation with tree-ring chronology and temperature showed a significant negative relationship during pre-monsoon, and April to May temperature, which is during monsoon season and is similar to previous studies in Pumijumnong, Eckstein and Sass [32] and Pumijumnong [21]. Correlation analysis revealed the importance of rainfall and moisture contrition, as opposed to temperature, to teak growth and climate relationship. Significant positive correlation of rainfall and moisture indicated the importance of moisture availability at the root zones. This indicated that increases in rainfall and moisture, typically during monsoon months, on an annual scale, influenced the variation in tree-ring width over the region. Therefore, based on these results, teak tree-ring chronologies can be used as a high-resolution proxy for past rainfall and moisture levels [40]. These chronologies have a positive correlation with rainfall and a negative correlation with temperature, indicating that increasing rainfall and decreasing temperature do affect the growth of teak trees. The season from April to May in the current year is a critical period of tree growth.

In this study, MT and UP chronologies could not be crossdated with each other. The relationship among these chronologies were found to be relatively low ($r = 0.33$, $n = 165$, $p < 0.01$). This result showed that growth of tree-ring structure in two different growth areas could be affected by a variety of non-climatic patterns among site variations, such as topography, nutrients, light, and internal factors (e.g., genetics, age, and hormones) [14,41]

For scPDSI, MT chronology showed a positive correlation with scPDSI from May to September and UP chronology shows significant positive correlation from January, and March to September, with the strongest being in May. May is also the transition period between the dry and wet season in the study area. This evidence is in agreement with the results of those of the Indian teak whereby the tree-ring variations had a significant positive correlation with scPDSI during different seasons, where positive scPDSI values represent wet conditions. The positive correlation between tree-ring chronology and scPDSI indicated the dependence of tree-growth on moisture [39] and is a clear indication of the effects of soil moisture as well as adverse dry conditions on teak growth [19,39].

### 4.2. Unstable Relationship between Tree-Ring Chronology and Niño Index

Rainfall across northern Thailand was correlated with the Indian summer monsoon (ISM), the Darwin pressure tendency (DPT) and the mean location of a subtropical ridge (SRT) [4]. Past records between Pacific sea surface temperatures (SSTs), more specifically, the El Niño–Southern Oscillation (ENSO), and summer monsoon rainfall of central and northern Thailand resulted in negative correlation [42]. Nevertheless, the values of the DMI showed no significant correlation with MT and UP chronologies. The central Pacific El Niño and the eastern Pacific El Niño (the Kaplan Niño 3, 3.4, and 4 during 1870–2016) exhibit discrete influences on monsoon rainfall over our study area; in particular, the relationship between tree-ring chronology and Niño index increased after the 1980s. Conversely, previous studies of tree-ring stable oxygen and carbon isotope in Merkus pines from Mae Hong Son province, northwestern Thailand [43,44], have shown that the relationship between temperature and rainfall in northwestern Thailand and Niño chronologies have disappeared since the 1980s. The spectrum of the summer monsoon in the central region of Thailand also showed a significant band of interannual/interdecadal timescales and variation that is linked to ENSO [42]. The relationship between Niño, SST, and ISM rainfall has limped since 1980, and the support for this result may be related to the south-eastward shift of the Walker circulation that broke down the relationship between ENSO and Indian monsoon and increased the frequency of the central Pacific ENSO [45].

## 5. Conclusions

Using the tree-ring chronology of teak (*Tectona grandis* L.f.) from the MT and UP sites in Tak province, northwestern Thailand, we have developed a 462-year MT chronology (1555–2016) that

would be the longest chronology of teak in Thailand. We have also developed a 162-year UP chronology (1855–2016). Two teak chronologies could not be crossdated. While it is obvious that the tree-ring chronologies contain low correlation ($r = 0.33$, $n = 165$, $p < 0.01$). Pearson correlation showed a high significant positive correlation with April to May rainfall during the first half of the monsoon season. Spatial correlation indicated that tree-ring chronologies could be representative of large-scale rainfall. However, further study of teak growth and climate relationship should be combined with other methods such as oxygen stable isotope from tree-ring cellulose that have a high potential for tracing the source of moisture and has been verified to represent a reliable hydroclimatic archive, to understand climate variability. This is particularly so with the dynamics of the Thailand monsoon, and there would need to be an expansion of teak chronology using teak stumps, historic buildings, and archaeological wood to examine the long-term Asian monsoon variability.

**Author Contributions:** Conceptualization, S.P., N.P. and P.P.; Methodology, S.P. and N.P.; Data curation, S.P.; Formal analysis, S.P., P.P. and S.B.; Investigation, S.P. and P.P.; Resources, N.P.; Validation, S.P. and N.P.; Writing-original Draft, S.P.; Writing-review & Editing, S.P., N.P., P.P. and S.B.

**Funding:** This research was funded by the Royal Golden Jubilee Ph.D. (RGJ-PHD) Program under the Thailand Research Fund (TRF) (Grant No. PHD/0076/2560) (For RGJ-PHD) and a project of Asian summer monsoon variability during the Holocene: A synthesis study on stalagmites and tree rings from Thailand and China by the Thailand Research Fund (TRF) (Grant No. RDG5930014).

**Acknowledgments:** We would like to thank forestry officers and co-workers for sample collection and preparation. Thanks to Kritsadapan Palakit, Faculty of Forest, Kasetsart University for encouragement and suggestion, Thomas Neal Stewart, Voravart Ratanadilok Na Bhuket and Kanokrat Buareal, Faculty of Environment and Resource Studies, Mahidol University for editing this manuscript and wonderful help.

**Conflicts of Interest:** The authors declare no conflict of interest.

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
