# Peer review of "Variation in Climate Signals in Teak Tree-Ring Chronologies in Two Different Growth Areas"

_forests, doi:10.3390/f9120772_

Round 1
Reviewer 1 Report
Tree ring chronologies in tropical regions are still rare and difficult to construct with proper statistics. The authors describe variation in climate signals in Tectona grandis between two sites in northwestern Thailand. They show that the two chronologies could not be crossdated, but both have a significant positive correlation with rainfall during pre-monsoon season. Additionally, the authors discuss the link between these chronologies and some teleconnections (ENSO/IOD). The study design setup, and analysis are very good, with sufficient sample size.
The manuscript itself, however, needs improvement:
Major comments on text:
- Several paragraphs are not where they belong:
Introduction: l41-l42. Limitations of the study are mentioned at the beginning of intro? This should be in the discussion.
Discussion: Oscillations, only in discussion you mention them. If this is a main message, discuss the graphs in the results and comment on them in the discussion.
Discussion L254-l267. Here you do not refer to own results, why not put this in introduction then? And also make shorter. Fig. 6 can be in results section.
Some questions on the content:
L51: gain moisture from PRECIPITATION from the Indian Ocean?
Table 1: could you add a column of average age of the trees? Also trees “taken over three times” rephrase. Also, healthy trees, what do you mean by that? Big crown?
101: “dendrochronological technique”? be more precize
Minor comments:
- Figures and tables:
Fig. 1 make more clear. Too many elements, consider putting some of the info in suppl. Material (not all climate diagrams should be included). The map in panel A should be simplified as well.
Fig. 5: label figure with panel a) and b)
Table 3: l180-l186 is the figure caption?
- English should be improved, several examples (this list is surely not complete, I did not mention the numerous smaller errors, which can be easily solved by the authors itself):
L12 living teak trees
L16 use sites instead of “areas”
L22 might be not affect
L47 Tree ring records
L48-49 rephrase
L56-57 rephrase
L135-136 rephrase
L230: which is during monsoon season
L248-l250 rephrase
In general: analysis is well performed and study set-up ok, and novel because of scarcity for these type of data.
Reviewer 2 Report
The article is very interesting and includes a lot of valuable information. Nevertheless, I have some remarks on the article. and concern mainly the section “Materials and methods”. It should be clear also to researchers not related directly to dendrochronology.
The remarks are also mentioned in “pdf”-file
Row 56/57
“This leads to the question of what role will the amount of rainfall play in the country?”
I believe that this sentence should be corrected and need an explanation. Maybe it's a translation problem.
Row 64
The title of the subsection, “Study Area and Climatology”
In my opinion, using here the word “climatology” is not correct. I’d like to suggest replacing the title with another one, e.g.
Only “Study area” or “Study sites” or “ Study area and characteristics of its climatic conditions”
Rows 79-81
“According to the Mae Sot Meteorological Station (AD 1951-2016) 79 in Tak province which is the nearest to the sampling site (MT and UP), the highest rainfall occurs in 80 August and the lowest rainfall occurs in December”
The maps (figure 1) shows that the station Mae Sot is the nearest station for MT, while in the case of UP the nearest meteorological station is Umphang – I kindly ask you to explain that
From the climate diagrams - Figure 1b - it follows that the annual course of thermal, rainfall and humidity conditions in MT and UP is similar, but the average values of precipitation and temperature in both places are different (Precipitation in Mae Sot: April -<25mm, may="">150 mm , June - ca 250, precipitation in Umphang: April ca 100mm, May 200, June<200mm, differences are also in average temperature values and relative humidity: That means, that the climatic conditions in both places are different, so there should be a description of climate conditions for both study sites- could you explain that?
Rows 85, 86
Figure 1. (a) Map of the study areas, Mae Tuen (MT) and Umphang (UP) (red circles) and the 85 meteorological stations (blue circles).
In my opinion there are errors in the figure 1 caption:
„ red circles” and „ blue circles” – in the figure 1 are other signatures (red triangles and dark circles)
Figure 1b-
In my opinion the scales on all climate diagrams should be the same, then you may see the climatic differences of these places. Of course, that has no influence on the results, but it gives a knowledge of the climatic conditions in this part of Thailand.
Figure 1c
I have a question for the authors. Is the Walter and Lieth Climate diagram for Umphang similar to the diagram from Mae Sot? if so, please provide information about that in the text, there is no need to present a new figure.
The subsection 2.3 and 2.4
I suggest expanding the method of transforming tree-ring data. It should be clear also to researchers not related directly to dendrochronology.
Rows 113-116
It seems to me that it would be good to give the source (literature) to the used ARSTAN program. E. G ARSTAN (Holmes, Dendrochronology Program Library, year) or website where the program is available.
Row 118
“These ring-width data were calculated using mean series inter-correlation (Rbar )” ?>>
I believe that this sentence should be corrected or explained. Did Rbar be used to standardization??
Maybe it's a translation problem.
Rows 120-124 -
Please explain, why the data from stations distant from the research area and from CRU were also used for the climate-growth analysis. What was the purpose of using this data as well?
In addition--: In Methods, there are no information…
1. There is no information about the time window for which the analyzes were performed. In “Results” in Figure 4 are the results that suggest using the time window - 12 month (January- December of current year)
2. There is no information how the spatial correlation with the CRU was carried out. It should be explained if it was performed by the men of each grid separately or it is the average of the whole area?
3. Authors don’t define the regional precipitation and the regional temperature in Methods. In rows 158 and 162 there are the terms of regional temperature and regional CTRU precipitation, but in the section Methods there is no explanation about the meaning of these terms and how these regional rainfall and regional temperatures were calculated
Results
Row 135
“ In this study, 112 ring-width MT series (67 trees) and 101 ring-width UP series (71 trees) of teak were constructed MT and UP tree-ring chronologies.
Please explain what was the reason for using just the part of samples/trees.
In MT site -there were took samples from 98 trees but just 67 were analysed
In Up site - there were took samples from 183 trees but just 71 were analysed
Why part of series was omitted from analysis? Low agreement with the rest of the dataset or different reason?
Rows 265-266
“Nevertheless the values of the DMI showed no significant correlation with MT and ….”
–there should be an explanation of abbreviation “DMI”

Round 2
Reviewer 2 Report
Thank you very much for including my comments and suggestions in the revised work. I'm sorry for the question about DMI, I did not notice this explanation.
Author Response
Dear reviewer
Thank you for giving us the opportunity to revise our manuscript. We highly appreciate your thoughtful and detailed comments and suggestions.
Round 3
Reviewer 1 Report
In my opinion, the article can be accepted without any further changes.
The article is well-organized, contain all of the components, and the sections are well-developed. The authors do a good job of synthesizing the literature. The methodology is clearly explained and he theory connect to the data. The article is well-written and easy to understand. The results of the research are clearly described, and the discussion carried out well. The responses to main aims of the research are clear too. In my opinion, there is an overall benefit to publishing this work. This article provides an advance towards the current knowledge and the authors have addressed an important long-standing question
The results of the Authors' research are valuable and
can be used for
further research on understanding the variability of climate in this
region of Asia, especially in order to examine the long‐term Asian
monsoon variability.The paper
has attracted attention from researchers and scholars specializing in
dendrochronology and dendroclimatology as well as in change climate.
Reviewer 2 Report
No further comments, except for making Fig. 1 clearer.